# A Scoping Review of Capacity-Building Efforts to Address Environmental Justice Concerns

**DOI:** 10.3390/ijerph17113765

**Published:** 2020-05-26

**Authors:** Dana H. Z. Williamson, Emma X. Yu, Candis M. Hunter, John A. Kaufman, Kelli Komro, Na’Taki Osborne Jelks, Dayna A. Johnson, Matthew O. Gribble, Michelle C. Kegler

**Affiliations:** 1Department of Behavioral, Social and Health Education Sciences, Rollins School of Public Health of Emory University, Atlanta, GA 30322, USA; kkomro@emory.edu (K.K.); mkegler@emory.edu (M.C.K.); 2Gangarosa Department of Environmental Health, Rollins School of Public Health of Emory University, Atlanta, GA 30322, USA; xinchun.yu@emory.edu (E.X.Y.); candis.m.hunter@gmail.com (C.M.H.); matt.gribble@emory.edu (M.O.G.); 3Department of Epidemiology, Rollins School of Public Health of Emory University, Atlanta, GA 30322, USA; john.alexander.kaufman@emory.edu (J.A.K.); dayna.johnson@emory.edu (D.A.J.); 4Environmental and Health Sciences Program, Spelman College, Atlanta, GA 30314, USA; nosborne@spelman.edu

**Keywords:** community capacity, environmental justice, community organizing, mobilization, social justice, advocacy

## Abstract

Environmental justice (EJ) efforts aimed at capacity building are essential to addressing environmental health disparities; however, limited attention has been given to describing these efforts. This study reports findings from a scoping review of community–academic partnerships and community-led efforts to address environmental inequities related to air, water, and land pollution in the United States. Literature published in peer-reviewed journals from January 1986 through March 2018 were included, and community capacity theory was applied as a framework for understanding the scope of capacity-building and community change strategies to address EJ concerns. Paired teams of independent analysts conducted a search for relevant articles (*n* = 8452 citations identified), filtered records for content abstraction and possible inclusion (*n* = 163) and characterized selected studies (*n* = 58). Most articles implemented activities that were aligned with community capacity dimensions of citizen participation (96.4%, *n* = 53), community power (78%, *n* = 45), leadership (78%, *n* = 45), and networks (81%, *n* = 47); few articles identified a direct policy change (22%, *n* = 13), and many articles discussed the policy implications of findings for future work (62%, *n* = 36). This review synthesizes three decades of efforts to reduce environmental inequities and identifies strategic approaches used for strengthening community capacity.

## 1. Introduction

As articulated by Dr. Robert Bullard and other prominent scholars and leaders in the environmental justice (EJ) movement, “everyone is entitled to equal protection and equal enforcement of our environmental health, housing, land use, transportation, energy and civil rights laws and regulations” [1], in all the places “where we live, work, play, worship, and go to school” [1]. In the United States, the EJ movement has grown in response to systematic inequities in exposures to lead and air pollution; groundwater contamination and drinking water safety; close proximity to noxious facilities and nuclear plants; location of landfills, incinerators, and abandoned toxic waste sites; placement of transportation thoroughfares; illegal dumping; superfund sites; and unequal enforcement of environmental laws [2,3,4,5,6,7]. Today, EJ also includes issues of immigrant and indigenous rights [8], climate change [9], green space [10], refurbishment of brownfields [11], and food justice [12].

The patterning of environmental inequity has received great attention and many studies have highlighted the pervasive nature of race as one of the most critically important predictors of exposure to environmental hazards [13,14,15]. Numerous research disciplines including American studies [16], economics [13], environmental health [5,6], epidemiology [4], sociology [1,17], philosophy [18], and politics [19,20] have documented discriminatory practices and the inequities of toxic waste distribution among low-income communities and communities of color. The explicit and implicit discrimination in environmental policymaking-the targeting of communities of color for toxic waste facilities, and the systematic under-resourcing and overburdening of marginalized communities—have been collectively termed environmental racism [21] and have plagued the United States for decades [3]. The disproportionate burden of environmental risks borne by communities of color necessitates the centering of environmental racism and inclusion of impacted communities in discourses about environmental injustice. Accordingly, prioritizing community concerns has been a focus for several foundation initiatives [22], as well as practitioners in the fields of public health, public policy, urban planning, economics, and social work. Community organizing and activism are foundational to the EJ movement and a core feature of much of the scholarship thereto pertaining [23,24,25,26,27,28].

The Principles of Environmental Justice, developed in 1991 by delegates of the First National People of Color Environmental Leadership Summit [29], stress the importance of focusing on “the rights to safe and healthy environments”, “equal partnership”, “mutual respect”, “cultural integrity”, and the “need for urban and rural ecological policies”. These Principles highlight the necessity of active community engagement in problem solving and decision making, in order to address the long-term solutions to environmental issues. Much of the academic literature on community health recognizes that inequalities in health are often a result of power relations that affect the distribution of resources and policy decisions [30]. Indeed, contesting the allocation of resources and demanding policy, social, and structural change requires the exercise of community power. Thus, building capacity in communities is vital for ameliorating health disparities, enhancing power [30,31], and redressing environmental injustice.

The practice of building capacity focuses on strengthening the characteristics that enable a community to identify, mobilize, and address social and public health problems [32]. However, the breadth of capacity-building efforts in the EJ sphere has not been thoroughly characterized in the academic literature, and we are aware of only six review articles have been published since 2002 to summarize various aspects of the literature [33,34,35,36,37,38]. They include: (1) an analytical review evaluating 44 EJ research studies on their scientific merit for purposes of policy and management decisions [33]; (2) a review of six National Institute of Environmental Health Sciences (NIEHS)/Environmental Protection Agency (EPA) funded Centers for Children’s Environmental Health and Disease Prevention Research implementing community-based participatory research (CBPR) to identify priority strategies to address environmental health concerns (i.e., establishing community trust, identification of community leaders, building on existing working relationships, assessment of community knowledge, and community involvement in development and implementation) [34]; (3) a systematic review assessing the influence of CBPR to address environmental health disparities and make community-level change [35]; (4) an in-depth analysis of four EJ related case studies representing CBPR partnership projects focusing on promoting sustainable environmental change and healthy public policy [36]; (5) a systematic review of EPA’s published funding opportunities from 1997 to 2013 for community-engaged research addressing environmental health and environmental inequities [37]; and (6) a review of action-oriented research highlighting six case studies addressing environmental justice and health issues with varying application of CBPR along the continuum of community engagement [38]. These reviews stressed the importance of community involvement and equitable collaboration in addressing environmental health concerns [34,35,36,37,38]; however, research translation into environmental and policy change were only narrowly discussed [33,34,35,36,37,38]. As a collective body, these reviews have made a significant contribution towards advancing EJ research, but there are still knowledge gaps related to the diverse scope of approaches employed at the community level to enhance capacity to address environmental issues.

In this review, we use *community capacity theory* as an organizing framework to better understand the strategies used to mobilize communities and address environmental problems [39]. Community capacity theory is an asset-based model comprised of 10 dimensions that underscore relationships, values and necessary resources for driving change and accomplishing desired goals. According to Goodman et al. [32], this theory offers a framework for understanding patterns for community building [32], and is comprised of the following dimensions: citizen participation, community power, community values, critical reflection, leadership, resources, sense of community, skills, social/organizational networks, and understanding community history. This theory has been applied to partnership development [36,40,41,42,43,44], program evaluation [45,46,47,48,49,50,51], and community organizing and policy advocacy [52,53]. In many ways, building community capacity is the “bottom-up” strategy of strengthening and developing skills, redistributing power, and increasing control over resources and decision making. The community capacity theory is particularly useful for analyzing EJ efforts, as it examines the structure and process of problem solving through an emphasis on community, social capital, competencies, and mobilization of resources needed for making and sustaining environmental change.

This review uses an expanded model of community capacity informed by the work of Freudenberg et al. [54], in which we have mapped specific capacity-building activities to each of the dimensions of community capacity for the purpose of responding to environmental threats. Freudenberg et al. [54] have also detailed general intervention strategies (i.e., authentic participation processes, CBPR, community organizing/social action, empowerment approaches, technical assistance, and training/technology transfer), that pertain to community ability to address environmental health hazards [54]. The utilization of these strategies has been shown to improve dimensions of capacity in multiple EJ-oriented CBPR interventions [36]. These strategies have also been influential in the development of health initiatives and community collaboratives [55], and advocacy for the removal of previously unrecognized community hazards [56]. This scoping review synthesizes three decades of capacity-building strategies at the community level to reduce environmental inequities related to air, land, and water pollution.

## 2. Materials and Methods

A scoping review, which seeks to outline the corpus of literature in a specific field of interest and explore research themes and gaps [57,58], was conducted to describe the literature published in peer-reviewed journals from January 1986 through March 2018 pertaining to community-engaged and community-led efforts aiming to address environmental threats, with a focus on undergirding theory and capacity-building strategy. Paired teams of research analysts participated in conducting this review. We used tailored search strategies to query Environment Complete, Political Science Complete, PubMed, SocINDEX, and Web of Science for peer-reviewed academic journal articles, using tailored searches described in Appendix A. Our search used text strings and controlled vocabulary in database-specific variations on the following terms: community capacity, community organizing, community mobilization, community health, community intervention, empowerment, environmental health, environmental justice, environmental policy change, environmental racism, equity, health disparities, program evaluation, and theory.

### 2.1. Historical Window

Trained research analysts searched for articles that detailed research published from January 1986 to March 2018. The year 1986 was chosen because this date immediately preceded the 1987 publication of the landmark “Toxic Wastes and Race in the United States” report, the first national study to examine the relationship between race and the treatment, storage, and disposal of hazardous waste [1,20]. This report catapulted the EJ movement and gave momentum to the subsequent organizing of the 1990 Conference on Race and Incidence of Environmental Hazards [59] and the 1991 First National People of Color Environmental Leadership Summit [60]. These meetings were soon followed by Executive Order 12898 [61], which established a federal inter-agency working group on environmental justice and mandated in 1994 that all United States federal agencies must, as part of their mission, attend in all their “programs, policies, and activities” to “disproportionately high and adverse health or environmental effects” on “minority populations and low-income populations” [61].

### 2.2. Language and Location

The database search was restricted to studies published in English that specifically detailed environmental justice and community-engaged environmental work conducted in the United States. Recognizing the value of literature published in other languages and countries, this review has a narrow focus on the United States due to the particulars of its history with regard to the racism and discriminatory practices central to environmental justice in the United States. While environmental equity is a concern broadly applicable across countries, U.S. governance and policy making processes are distinct.

### 2.3. Eligibility

Studies were considered eligible if: (1) the study focus was related to mobilization, empowerment, and action around issues of air, land, and water pollution; (2) the specific environmental concern was related to brownfields [62], drinking/water quality, air or water pollutant emissions, groundwater contamination, illegal dumping, incinerators, landfills, lead poisoning, proximity to noxious facilities and nuclear plants, Superfund sites [63], or toxic waste; and (3) the study authors identified the target community as low-income, marginalized, disenfranchised, minority, or a community of color. Articles were considered eligible for this scoping review if they used any variation of community-engaged qualitative or quantitative methods, and included: (1) a resolution or reduction of an environmental health concern, (2) the implementation of strategy to address environmental health disparities, or (3) a strategy for enhancing community capacity, empowerment, leadership or decision making in relation to environmental concerns.

### 2.4. Data Abstraction

During a two-week period of time, a search query was conducted and a total of 8452 unique citations were identified from five databases: PubMed (searched 19 March 2018); Web of Science (searched 22 March 2018); SocINDEX (searched 25 March 2018); Environment Complete (searched 28 March 2018); and Political Science Complete (searched 31 March 2018). Citations were subsequently imported into a shared EndNote reference management file. Over the course of two months, articles were evaluated and sorted by a team of two analysts for relevance based on title and abstract (see Figure 1). Any manuscript considered possibly relevant by either analyst was carried forward for full text review (*n* = 925). Within an additional two-month period of time, based on the inclusion criteria aforementioned, the full text review of articles was carried out by two analysts using an online Google Form (included abstraction measures are detailed in Section 2.5). Using this online screening form, a total of 762 articles were excluded from content abstraction for one of the following nine reasons pertaining to the research methodology, study design, or study type:

(1) there is community involvement but no action, strategy or policy change either planned or enacted; the a study lacks community mobilization; (2) it is a conference abstract, book review, or other review type; (3) the research described is incomplete (i.e., protocol registration, cohort announcement, pilot study); (4) the publication is grey literature (e.g., editorial, magazine, newsletter, newspaper, speech, website); (5) the research does not directly engage with community (e.g., school-based only, risk assessment); (6) the research is not related to pollution (i.e., focus on physical activity, obesity, nutrition, greenspace, tobacco); (7) the research is presented as a summary report or government document; (8) the research is a survey only (e.g., no report of partnership or organizing to address EJ concern); or (9) excluded for other reasons (i.e., commentary, international study, non-English language, non-human research, non-point source pollution).

Upon completion of screening, all documents marked for possible inclusion were retained for a deeper level of filtering following content abstraction (*n* = 163). A second Google Form was created to allow for data abstraction, and discrepancies within the team related to inclusion were resolved by consensus, resulting in a final number of 58 articles for inclusion in this analysis (see Appendix A). Data were imported into SPSS Version 24 (IBM, New York, NY, USA) for descriptive analyses [64].

### 2.5. Measures

Data from relevant manuscripts were abstracted using a standardized form to identify the policy, system, or environmental (PSE) change target [65,66]; theoretical framework cited; activities to enhance community capacities as categorized by theoretical domains; as well as general capacity-building and community change strategies. Additional abstraction measures (further detailed in results) include author affiliation, research design, study setting and population demographics (see Appendix A for Word version of Google Form used for data abstraction).

#### 2.5.1. Author Affiliation

Author affiliation was classified based on the first author of the included publication and categorized as being affiliated with an academic institution; with a community based (CBO) or non-profit organization (NPO); with a foundation (e.g., an endowed entity, institute, or foundation); or as having unknown affiliation (i.e., the author did not provide an affiliation).

#### 2.5.2. Research Design

Studies were categorized into one of six research designs: (1) a case study, (2) an evaluation study, (3) an observational/cross-sectional study, (4) an observational/longitudinal study, (5) a mixed methods design, or (6) a natural experiment (see Appendix A, Data Abstraction form for detailed study design definitions).

#### 2.5.3. Source of Funding and Community Demographics

The ‘source of study funding was classified based on the entity providing financial support for the research: a foundation or nonprofit; the NIH/NIEHS; another government funding source [e.g., the U.S. Environmental Protection agency (EPA), the U.S. Centers for Disease Control and Prevention (CDC)]; or an unspecified source of funding.

The ‘community demographics’ variable classified studies by the reported race/ethnicity of the community (i.e., African American, American Indian/Alaskan Native, Hispanic/Latinx, White, multiple ethnicities, or without an explicit description provided), as well as whether the community was described as low income, impoverished, or underserved.

#### 2.5.4. Targeted Pollution Concern

Identified pollution concerns were categorized as: air pollution/air quality concerns; illegal dumping; hazardous waste inclusive of brownfields, Superfund, chemical contaminants, soil contaminants, fish contaminants; and/or water quality concerns related to drinking water or groundwater.

#### 2.5.5. Assessment of Policy, System or Environmental (PSE) Outcomes

PSE outcomes included: reduction/clean-up/remediation; mitigation of pollution; enforcement of laws or review of permits or increased compliance; prevention of industrial development; and/or legislation to address pollution. Failed policy outcomes were coded separately as failures.

#### 2.5.6. Theoretical Framework

Studies were categorized as using theory if they mentioned, identified, or used any reference to a social science theory, model or framework. If a theory was used it was identified and its study application was categorized as one of the following: (1) informed data collection instrument; (2) informed sampling methods; (3) identified constructs as a guiding framework for analysis; (4) developed as a result of analysis (e.g., grounded theory); (5) mentioned in the introduction/discussion but not operationalized or measured; or (6) there was no mention of a guiding theory.

#### 2.5.7. General Strategies to Enhance Community Capacity

Our classification of strategies for organizing around environmental pollution concerns were informed by Freudenberg’s intervention strategies to increase community capacity to protect against environmental stressors [54]. In this review, we considered six potentially overlapping capacity-building strategies: (1) authentic participation; (2) CBPR; (3) community organizing and social action; (4) empowerment approaches; (5) technical assistance; and (6) training and technology transfer. These strategies were specifically chosen to consider varying ways in which capacity can be addressed to shape PSE change.

#### 2.5.8. Direct Community Change Strategies

In addition to these capacity-building strategies, the following strategies were author identified through an iterative process of reviewing EJ literature: civil disobedience, letter writing, litigation, media advocacy, partnership, coalition building, and policy advocacy (see Table 1 for measurement definitions and examples). Given the variation in study design, these additional strategies were important to include as many approaches identified in this list did not fall within traditional academic-led research practices.

#### 2.5.9. Activities to Strengthen Specific Dimensions of Community Capacity

The community capacity theoretical framework operationalizes aspects of social agency across individual, organizations and networks suggesting that capacity building, identifying local concerns, and encouraging engagement builds collective power to mobilize and address concerns [32,67]. Our analysis considered 10 categories of activities (see Table 2) reflecting theoretical dimensions of community capacity. This list of activities was adapted from the work of Freudenberg et al. [54] and lists a distinct set of particular actions that may have been undertaken to enhance capacity to address environmental issues.

## 3. Results

A total of 8452 unique articles were identified by our search (Figure 1), of which 925 articles passed title-abstract review, 163 during full text review were considered to merit further evaluation (e.g., content extraction), and 58 relevant articles were identified as satisfying all inclusion criteria. The most common reasons for exclusion at the last stage of evaluation (*n* = 163 to *n* = 58) were that the article was a survey assessing the environmental concern without translation of findings to the target community; or the article did not describe how the study findings were used to assist community in making environmental change.

### 3.1. Description of Study Population and Setting

As shown in Table 3, the articles we found that satisfied our inclusion criteria were predominantly authored by academics across varying disciplines (84%, *n* = 49) followed by a small percentage comprised of community-based entities and foundations (10.3%, *n* = 6) and others (5.1%, *n* = 3). Many authors also described the communities in which their studies were conducted as being predominantly African American (38%, *n* = 22), Latinx (29%, *n* = 11), or did not describe a specific target community’s race or ethnicity (24%, *n* = 14).

### 3.2. Study Design

The majority of the articles we identified that satisfied inclusion criteria were case studies (70.7%, *n* = 41), followed by evaluation studies (10.3%, *n* = 6) and mixed methods research (10.3%, *n* = 6); with only a few examples of cross-sectional (5.2%, *n* = 3), experimental (1.7% *n* = 1), or longitudinal (1.7%, *n* = 1) studies.

### 3.3. Targeted Pollution Concerns

The two most common environmental priorities described in the studies were air quality (41.4%, *n* = 24; Table 3); and hazardous chemical exposures, inclusive of concerns related to the presence of brownfields, Superfund sites, toxic chemicals, soil contamination, or water contamination (24.1%, *n* = 14). We found fewer studies focused on illegal dumping (1.7%, *n* = 1) or water quality alone (1.7%, *n* = 1). Close to one-third (31.0%, *n* = 18) of the studies focused on multiple pollution issues.

### 3.4. Environmental and Policy-Related Outcomes Resulting from Advocacy Efforts

Many of the studies we identified did not result in any environmental change (60.3%, *n* = 35) or a direct policy change (24.1%, *n* = 14; Table 3). Many authors discussed the policy implications of their findings for future work (62.1%, *n* = 36). More than one-third of the studies we identified as satisfying inclusion criteria detailed having an environmental impact in some regard (36.2%, *n* = 21), such as the reduction of exposure to an environmental pollutant, the cleanup or remediation of pollution concerns, or another environmental effort (e.g., roadway clearing, relocation, or establishing a monitoring station). Regarding policy-related outcomes, about one-third of studies (29%, *n* = 17) reported mitigating the environmental concern by reducing the risk of the community to the exposure of the environmental pollutant (e.g., technical modifications in plant operations, update monitoring systems, or reduce emissions). We identified fewer examples of legislative resolutions to address environmental concerns (22.4%, *n* = 13) or successful prevention of industrial development (20.7%, *n* = 12). A small percentage (18.9%, *n* = 11) reported being able to encourage enforcement of an existing environmental law, regulation, or review of a conditional permit). Fewer studies (5.2%, *n* = 3) discussed other policy-related outcomes such as implementation of new local policies (e.g., stop signs, bus idling, or implementation of new air regulations) and several studies discussed failed advocacy efforts in making policy change (*n* = 17.2%, *n* = 10).

### 3.5. Theoretical Frameworks

More than half of the studies we identified reported using a theory, model, or framework (51.7%, *n* = 30). Among these, the most popular mention of theory was related to community capacity (22.4%, *n* = 13) followed by an environmental justice framework (12.1%, *n* = 7). When applied, theory was commonly used in the introduction or discussion as a reference point for framing of the central issue (19%, *n* = 11), without being operationalized or measured. Additional application of theory was seen in the development of constructs for analysis (13.8%, *n* = 8), informing data collection measures (8.6%, *n* = 5), or informing sampling methods (5.2%, *n* = 3).

### 3.6. General Strategies to Enhance Community Capacity to Address Pollution Concerns

A wide variety of capacity-building strategies were described across included articles (see Table 3 data). The most frequently identified strategies included the presence of authentic involvement and participation of the community in planning and data collection (96.4%, *n* = 53); the implementation of empowerment approaches (77.6%, *n* = 45); and community organizing/social action (58.6%, *n* = 34) that encouraged people to advocate for themselves and make demands for increased resources. CBPR was also a common strategy (50%, *n* = 29) in which community members participated at varying levels in the selection of issues, study design, interpretation of findings, and dissemination (e.g., lay heath advisors and community advisory board; see Appendix A) [47,68,69,70,71,72,73,74,75,76,77].

### 3.7. Direct Community Change Strategies

Other community change strategies described for enhancing capacity to address pollution concerns (see Appendix A) included media advocacy (32.8%, *n* = 19), litigation (31%, *n* = 18), and policy advocacy (24.1%, *n* = 14) with less frequent mentions of citizen science activities (22.4%, *n* = 13), civil disobedience (17.2%, *n* = 10), photovoice (12.1%, *n* = 7), and letter writing (10.3%, *n* = 6).

### 3.8. Activities to Strengthen Specific Dimensions of Community Capacity

While we found the concept of community capacity only explicitly discussed in a few articles [47,68,69,78,79,80,81,82,83,84,85,86], many studies implemented activities that were aligned with dimensions of community capacity theory (see Appendix A). Among included articles, several discussed activities related to understanding community history (37.9%, *n* = 22), displays of critical reflection (22.4%, *n* = 13), identification of community values (18.9%, *n* = 11), and implementation of skills development (20.7%, *n* = 12). About two-thirds of included articles discussed activities related to establishing a sense of community (62.1%, *n* = 36) and the identification of resources (58.2%, *n* = 32). The most discussed activities in our sample of studies included citizen participation (96.4%, *n* = 53), community power (77.6%, *n* = 45), leadership (77.6, *n* = 45), and networks (81%, *n* = 47). Examples are detailed below that represent these most commonly cited dimensions.

#### 3.8.1. Citizen Participation

In the context of community capacity theory, citizen participation entails the involvement of a diverse group sharing different interests and perspectives for defining a problem and taking collective action [32]. When the dimension of citizen participation was discussed in the articles we identified as satisfying inclusion criteria, activities entailed the recruitment of residents as study participants, offering incentives, conducting outreach activities, neighborhood canvasing, or formal/informal community forums to discuss research intentions, recruit participants or share study findings [47,68,69,70,71,72,73,74,75,76,77,79,80,81,82,83,84,85,86,87,88,89,90,91,92,93,94,95,96,97,98,99,100,101,102,103,104,105,106,107,108,109,110,111,112,113,114,115,116,117,118,119]. Specific examples include the work of Cohen et al. [79], Loh et al. [108], and Green et al. [100]. The article by Cohen et al. [79] discusses the application of community-driven hypothesis generation/testing, the training of community surveyors, and collection of environmental health survey data to document and quantify neighborhood concerns. The article by Loh et al. [109] describes community-based research efforts that entail the training of residents to monitor and collect particulate matter samples to document air pollution “hot spots” contributing to high rates of asthma. The article by Green et al. [100] details the implementation of a comprehensive community-based environmental risk assessment involving the facilitation of focus groups to assess the relevancy of strategic initiatives and social action materials.

#### 3.8.2. Community Power

In the context of community capacity theory, community power represents the ability to create or resist and be influential with respect to changing conditions [32]. When the dimension of community power was identified in the studies we identified as satisfying our inclusion criteria, residents/community members most often used the strength of scientific data to mobilize and address their environmental concerns [47,68,69,70,71,72,74,75,78,89,90,91,92,93,94,95,96,98,99,101,102,103,104,108,111,112,113,115,116,117,118,119,120]. One example is the work of Bell [89]; in her description of a photovoice project which impacted residents of a coal-mining Appalachian community mobilized through photo-storytelling and shared experiences, she exhibited photo stories for community discussion, and used this content to develop a legislative document for educating about coal slurry injections and impoundments. Community power is also represented in the work of Jiao et al. [106] through the use of public participatory geographical information systems (PPGIS), in which researchers developed a risk communication tool that empowered residents by allowing them to educate themselves on the potential risks from industrial sources in close proximity to their community.

#### 3.8.3. Leadership

The dimension of leadership in community capacity theory represents the ability of a person or group to direct and encourage participation, and the sharing of information and resources among a network of community participants [32]. Several of our identified studies displayed leadership development through advocacy education, training, and assistance with strategic planning and policy development [47,68,69,71,72,74,76,77,78,80,82,84,85,86,87,89,90,91,92,93,94,95,96,97,98,99,100,101,102,103,104,105,106,107,108,112,114,117,118,121,122,123]. In the Theater of the Oppressed, described by Sullivan et al. [115], both community power and leadership development are illustrated. They describe a unique community environmental forum that used image theater and improvisation to teach environmental toxicology. Through improvisational theater, participants developed a working knowledge of risk assessment, learned to apply analytic tools, and were taught how to frame community dialogue and understanding about environmental injustices and regulatory policy issues. A CBPR study, conducted by Balazs [88], further exemplifies the dimension of leadership through intentional community co-ownership efforts implemented in the research process. In this example, community members were trained to conduct indoor/outdoor air monitoring, were involved in collective discussions/negotiations of study methodologies and participated in the development of a protocol for the reporting back of research findings. The work of Wier et al. [117] also displays leadership development in the context of traffic/transportation pollution and the subsequent health burdens experienced by neighborhood residents. This study created opportunity for the connection of community knowledge, research, and community action through the facilitation of community workshops, educational theater skits, and development of pamphlets with community stories and comic art to educate about health effects and the need for action [117].

#### 3.8.4. Networks

The community capacity theory dimension of social and organizational networks encompasses the supportive interactions between groups of people and organizations, formation of new partnerships, and shared decision making [32]. Partnership and networking activities included the intentional involvement and collaboration with concerned citizens, environmental activists, and/or other health professionals for environmental health promotion activities, as well as research planning and sharing of data with community and dissemination of research findings [47,68,69,70,72,73,74,75,76,77,78,79,80,83,84,85,86,87,91,92,93,94,96,97,98,99,101,102,103,104,105,106,107,111,112,113,114,115,116,117,120,121,123]. Allen et al. describes a diverse coalition comprised of the sovereign Lummi Nation, environmental organizations, faith-based entities, and other citizen groups in the development of a people’s initiative [70]. This coalition focused on getting anti-coal politicians elected to local government, use of federal/local regulatory permitting processes to influence equitable outcomes, and provided information to the local community about the negative impacts of fossil fuel on local health, ecosystems and culture. Also using CBPR principles, Haynes et al. [102] and Sadd et al. [82] created academic–community partnerships to jointly shape their environmental health research. Appalachian residents who were exposed to airborne manganese comprised the community advisory board and shared their perceptions about air quality, health concerns, and risk communication [102]. Additionally, local Los Angeles residents collaborated with researchers to develop an EJ screening method that gathers data about pollution sources and maps the proximity to identify the clustering of hazardous facilities [82].

## 4. Discussion

This scoping review of capacity-building efforts to address EJ concerns identified a variety of capacity-building strategies at the community level, including community-engaged partnerships and community-led efforts that have been employed to address pollution-related environmental inequities. We have focused on understanding the presence of capacity dimensions and subsequent efforts used to enhance a community’s capacity, as these are fundamental within the context of EJ advocacy in promoting solidarity and generating power to overcome inequities. The ultimate goal of building capacity and cultivating partnerships is to address problems, which often requires enacting meaningful change at the system, structural, and policy levels. However, these types of change are often shaped by local context and may require more time than the span of any research study. Therefore, we consider a range of strategies discussed across published studies as potentially relevant for EJ progress, whether or not these studies explicitly describe impactful PSE change.

### 4.1. Community Capacity as a Theoretical Lens to Address EJ

There were numerous studies that identified and incorporated some type of social science theoretical model for informing data collection, sampling methods, and/or framing study findings within an EJ context [70,78,80,86,91,92,93,96,101,121,122]. Some of the articles frame their work in relation to the advocacy coalition framework [70,124], the collaborative problem-solving model [78,86,125], empowerment theory [91,126], resource mobilization theory [92,96,127], and the environmental justice framework [71,80,91,96,101,121,122,128]. A common theme across these models, as discussed in the studies, was the intentional centering of marginalized populations, a focus on the disproportionate impacts of toxic environmental exposures, and an emphasis on advocacy. According to Bullard and Johnson, this framing is most essential in the development of strategies “to eliminate unfair, unjust, and inequitable conditions and decisions” [128], and it is critical to have specific activities for invoking and sustaining environmental change. Community capacity theory is not only in alignment with this environmental justice framing, but the theoretical tenets provide a structure for developing activities needed for driving and sustaining environmental change. The core attributes of community capacity theory focus on strengthening and developing skills, redistributing power, increasing control over resources, and increasing decision-making ability [32], all of which are embedded within EJ thinking. Accordingly, because of community capacity theory’s emphasis on community competencies, social capital, partnership, and mobilization of resources, community capacity theory is a useful and valuable framework for understanding the process and structure of EJ advocacy research.

In the application of community capacity theory within the context of EJ advocacy, we used the Freudenberg et al. [54] expanded model in which specific capacity-building activities were mapped to each of the 10 dimensions for the purpose of responding to environmental threats. We regard these dimension-related activities as falling into two distinct sets. The first set of activities was commonplace in included studies and focused on offering incentives (participation); educating about environmental issues (power and leadership); embedding research in existing community structures (sense of community); and creating partnership opportunities (networks).

In contrast, a second set of activities less frequently documented in these case studies was explicitly focused on considerations for effecting change. About half of the articles highlighted activities related to social and financial capital (resource development), while even fewer articles highlighted capacity-building strategies involving reflection on past community successes or limitations (critical reflection). Similarly, there was little attention given to activities related to strengthening the understanding of root causes (community history); offering of technical assistance or skills development opportunity (skills development); or alignment of the research with community values (community values).

Across these observations it is important to note that all dimensions are viewed as instrumental for building capacity; however, many of the dimensions ascribed to the first set are comprised of activities that are more superficial: for example, engagement was often limited to mere study participation. This sort of limited approach to community participation tends to emphasize academic priorities without emphasizing the practices and activities that are necessary to foster community buy-in, shared understanding, community cohesion, or program sustainability. In contrast, the second set of capacity-building activities were utilized less often yet they may be the most critical for deeper engagement and prioritization of community-driven needs and values. While likely necessary for establishing trusting relationships, understanding context, aligning with community priorities, and supporting community development, this second set of activities also require greater intention and investment.

### 4.2. Observations Concerning EJ Community Research Practices

This review highlights some of the breadth of the community capacity-building strategies employed in pollution-related EJ efforts, as documented in peer-reviewed journals, and is aligned with scholarship on the importance of building capacity [46,47,52,53,54,129,130,131,132]. In our sample of studies, it was less common to see an emphasis on authentic engagement, shared priorities, community development, and sustainability with respect to addressing environmental concerns. Few articles described how to strengthen capacity beyond the “surface level” activities of offering incentives, providing education, aligning with existing community structures, and creating partnership opportunities. We encourage EJ research efforts to consider deeper capacity-building strategies with potential for collaboration that empowers communities to potentially overcome environmental inequities.

### 4.3. Limitations

This review was systematic in identifying peer-academic reviewed literature on community EJ research conducted in the United States, published in English. As such, it paints a clear picture of academic EJ community practice over several decades, but it may not be representative of the large swath of the EJ capacity-building efforts not documented in peer-reviewed academic journals. The academic literature alone is inadequate for fully characterizing which capacity-building strategies are most effective in EJ. The modest reporting of PSE change outcomes within the sample of identified studies unfortunately precludes comparative effectiveness claims about which community capacity-building strategies are more likely to lead to desired PSE change even within the space of academic-community partnerships. Even if ultimate PSE change outcomes were known, it would be challenging to make synthesizing inferences from this literature, as varying methodologies are used across studies, research designs are incomparable, the application of the term capacity and the related strategies are inconsistent across studies, measurement tools varied across studies, and there is great variation in community-specific challenges.

## 5. Conclusions

This review provides an overview of activities for building capacity and working toward environmental community change; and to our knowledge is the first review of its kind to synthesize a myriad of community-led and community–academic partnership activities in this way. One of the major strengths of this review is the inclusion of multiple databases being reviewed by two independent analysts, and the application of a community capacity theoretical framework. The asset-based model of community capacity that we applied in this review clearly distinguishes 10 dimensions and offers one approach for operationalizing strategies with these dimensions. We have identified several examples of deep community capacity-building strategies that may benefit the EJ research community. Through clear and intentional application of a theoretical framework, we can systematically identify common strategies for building capacity and begin to identify research questions related to how best to strengthen community capacity to address EJ concerns. Accordingly, it is our intention that the findings of this scoping review can aid in improving the practice of community-engaged EJ research and enable future research that gives greater insight on what makes for transformative change.

## Figures and Tables

**Figure 1 ijerph-17-03765-f001:**
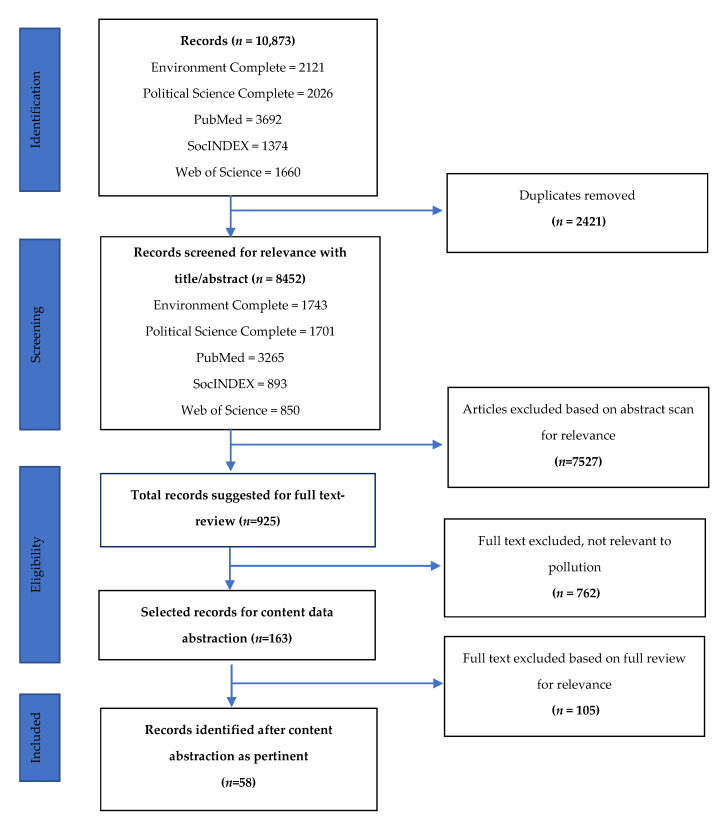
PRISMA (Preferred Reporting Items for Systematic Reviews and Meta-Analyses) Flow Diagram.

**Table 1 ijerph-17-03765-t001:** Strategies to enhance community capacity.

General Strategies to Enhance Capacity Derived from Freudenberg [54]	
**Authentic Participation Processes**	Agency designed participation processes that improve community capacity by getting people involved early, providing them with information and resources for full participation, and ensuring that outcomes reflect their participation.
**Community-based Participatory Research (CBPR)**	A research process in which community residents participate in selecting issues, designing studies, interpreting findings, and presenting results to policymakers for the purpose of reducing environmental health inequities and promoting healthier public policies (e.g., citizen science practices and utilization of the lay health advisor training model in which individuals of the community are trained as resources to assist in the education and distribution of materials for research studies)
**Community Organizing/Social Action**	Community mobilization and organization to enable a disadvantaged segment of the population to make demands on the larger community for increased resources and more equitable policies.
**Empowerment Approaches**	Process by which individuals, communities, and organizations gain power and mastery over their lives in the context of changing their social and political environment to improve equity and quality of life.
**Technical Assistance**	Tailored support that enables community participants to gain information or skills to solve problems or to participate more effectively in decision-making processes.
**Training and Technology Transfer**	Process by which community participants gain knowledge, skills, competencies, or technologies that enable them to participate in assessing and remediating environmental hazards and participating in relevant policy deliberations.
**Community Change Strategies (author-created)**	
**Civil Disobedience**	The refusal to comply with certain laws or to pay taxes and fines, as a peaceful form of political protest, that often includes nonviolent techniques such as boycotting, picketing
**Letter Writing**	An organized effort to coordinate as many people as possible to write to a decision maker (legislative or facility) asking them to take a particular action.
**Litigation**	The process of taking legal action to enforce of defend a legal right.
**Media Advocacy**	Strategic use of traditional or social media outlets to disseminate information and promote policy initiatives.
**Photovoice**	A participatory method that has community participants use photography, and stories about their photographs, to identify and represent issues of importance to them.
**Policy Advocacy**	Analysis of the cause of the problem and development of policy-based solutions to create sustainable change.

**Table 2 ijerph-17-03765-t002:** Activities to strengthen dimensions of community capacity.

Activities to Strengthen Dimensions of Community Capacity *	
**Citizen Participation**	Incentives for participation are offered; outreach is conducted to uninvolved sectors of population; door-to-door canvasing; and conducting community forums to bring formal and informal community leaders together to consider environmental health issues.
**Community History**	Context and analysis of previous efforts are described, assisting residents to study and analyze previous health and environmental issues facing community; preparation of reports aimed at community residents to develop understanding relative to history.
**Community Power**	Empowerment or building of power is explicitly stated; providing community with information so they can confront special interests effectively; supporting political reforms that level the playing field for those with less influence; providing scientific information that can be used in political arena.
**Community Values**	Values, shared norms and standards that underlie public health efforts related to environment, social justice, and democracy are articulated; research is described within context of defending community values against disease promoting entities.
**Critical Reflection**	Identification of successes and limitations of actions, assisting community residents to analyze and reflect on successes and limitations of their actions to promote environmental health.
**Leadership**	Preparation of environmental activists to be leaders; education of community leaders about environmental issues; assistance provided with strategic planning and policy development; and leadership development or training is explicitly stated.
**Resources**	Research described as being a bridge between community and external resources (e.g., state health dept, foundations); assisting participants to identify local assets; assistance provided with writing grants and/or working with funders to support community groups; use of grant funding to assist with community project.
**Networks**	Partnership explicitly stated; description of work as being a support structure for nurturing local, regional, and/or national coalitions that bring together concerned citizens, environmental activists, scientists, health professionals, and others for environmental health promotion activities.
**Sense of Community**	Support for community events (pre-existing or community-led events) that build sense of identity; research details the creation of a safe space/forum for community residents to discuss, analyze, and study environmental health issues.
**Skills**	Offering workshops and technical assistance on environmental health issues; creation of opportunities for participants to exchange skills; intentional effort to link skills inside and outside community to those with needs.

* Activities detailed are derived from Freudenberg et al. Public Health Strategies to Build Community Capacity for Environmental Health Action [54].

**Table 3 ijerph-17-03765-t003:** Review Article Characteristics, *n* = 58.

	*n*	%
Author Affiliation and Discipline		
**Academic Discipline**	49	84.5
**Community-based (CBO), Non-profit (NPO) Organizations**	3	5.2
**Foundation, Institute**	3	5.2
**Information not Provided**	3	5.2
Source of Study Funding		
**Foundation or Nonprofit Support**	20	34.5
**NIH/NIEHS Funding**	17	29.3
**Other Government Funding Source (CDC, EPA)**	5	8.6
**Not Specified**	15	25.9
Target Community		
**African American**	22	37.9
**American Indian/Alaskan Native**	3	5.2
**Hispanic/Latinx**	11	28.9
**White**	4	6.9
**Multi-ethnic**	4	6.9
**No Explicit Description Provided**	14	24.1
Community Description		
**Low-income, Impoverished or Underserved**	34	58.6
Research Design		
**Case Study**	41	70.7
**Evaluation**	6	10.3
**Mixed Methods**	6	10.3
**Natural Experiment 1 Measurement Point**	1	1.7
**Obervational and Longitudinal**	1	1.7
**Observational Cross-sectional**	3	5.2
Pollution Focus of Research Study		
**Air Pollution/Air Quality**	24	41.4
**Hazardous Waste (Brownfield, Superfund)**	14	24.1
**Illegal Dumping**	1	1.7
**Water Quality (Drinking or Groundwater)**	1	1.7
**More than one Pollution Focus**	18	31.0
Theoretical Framework *		
**Environmental Justice Framework**	7	12.1
**Community Capacity Theory**	13	22.4
**Other Social Theories or Frameworks**	19	32.8
Application of Theory *		
**Informed Data Collection Instrument**	5	8.6
**Informed Sampling Methods**	3	5.2
**Constructs used for Analysis**	8	13.8
**Mentioned but not Operationalized or Measured**	11	19.0
**No use of Theory Reported**	28	48.3
Capacity-building Strategies *		
**Authentic Participation Processes**	53	96.4
**Community-based Participatory Research (CBPR)**	29	50.0
**Community Organizing/Social Action**	34	58.6
**Empowerment Approaches**	45	77.6
**Technical Assistance**	14	24.1
**Training and Technoogy Transfer**	12	20.7
Community Change Strategies *		
**Citizen Science**	13	22.4
**Civil Disobedience**	10	17.2
**Letter Writing**	6	10.3
**Litigation**	18	31.0
**Media Advocacy**	19	32.8
**Photovoice**	7	12.1
**Policy Advocacy**	14	24.1
Observed Environmental Outcome		
**Clean-up of Pollution Concern, Reduced** **Exposure, Remediation of Toxic Waste**	18	31.0
**Increased Regulationof PM_2.5_**	0	0
**Other**	3	5.2
**None Reported**	35	60.3
Policy Related Outcomes as a Result of Advocacy Efforts *		
**Enforcement Environmental Law/Regulation; Review of Conditional-use Permit**	11	18.9
**Increased Compliance; Mandatory Payment of Fines for Pollution and/or Safety Violations**	0	0
**Legislative Resolution to Address Toxic Emissions**	13	22.4
**Mitigation of Concern**	17	29.3
**Prevention of Industrial Development of Noxious Facility**	12	20.7
**Other Policy-related Outcomes**	3	5.2
**Application of Any Aforementioned Advocacy Efforts with Unsuccessful Policy-related Outcomes**	10	17.2
**No Policy-related Outcome Reported**	14	24.1
**Mention of Policy Implications of Research Findings**	36	62.1

Note: * Categories not mutually exclusive.

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
