# Peer review of "A Scoping Review of Capacity-Building Efforts to Address Environmental Justice Concerns"

_ijerph, 2020, doi:10.3390/ijerph17113765_

Round 1

Reviewer 1 Report

Congratulations for your study. Very relevant and important results.

The paper is well organized and the methodology well described. 

Some suggestions:

  • Maybe you can improve the quality of the images:

PRISMA Flow Diagram is a litle distortion and should be improve (line 191/192)

  • Suggest also you describe the timeline of the research phases.

How long did you take to analyse  8,452 papers (even if just title and abstract)? How did the research team organized it self to respond to this phases?

  • Page 20 should be in portrait orientation.
  • Pag. 32 to 34 should be in landscape oritentation.
  • The printscreen of pág 37 has a "Ctrl" simbol above the text.

Author Response

Thank you for the suggested edits, please see the attachment.

Reviewer 2 Report

The manuscript is excellent as a review and to summarize decades of efforts to reduce environmental inequities and for the identification of approaches used for strengthening community capacity. However, it is very extensive and could summarize the results better and have a summarized methodology.

How the study focuses on discuss the necessity of community organizing and collective problem solving, some articles can be used to discuss the results, such as:

https://doi.org/10.3390/ijerph16071118

https://doi.org/10.3390/su12041682

https://doi.org/10.3390/f8040116

https://doi.org/10.3390/su71114537

Another suggestions:

  1. improve the quality of figure 1;
  2. make table 4 as supplementary material;
  3. need to explain better why you organized / chose the items in tables 1 and 2;
  4. I suggest presenting the results separating by journals, year, main authors and etc.

Author Response

(The authors gave the same response as above.)

Round 2

Reviewer 2 Report

No suggestions for authors.